# BUDGET-AWARE TEST-TIME SCALING VIA DISCRIMINATIVE VERIFICATION

## ABSTRACT

Test-time scaling is a powerful strategy for boosting the performance of large language models on complex reasoning tasks. While state-of-the-art approaches often employ generative verifiers to select the best solution from a pool of candidates, this method incurs prohibitive computational costs, limiting its practicality. In this work, we pivot the focus to a more budget-aware paradigm: discriminative verification. We conduct a thorough empirical analysis and demonstrate that while discriminative verifiers may underperform in isolation, combining them with self-consistency in a hybrid approach creates a powerful and efficient ~~selection~~test-time scaling mechanism. ~~These hybrid methods consistently outperform self-consistency with negligible computational overhead (e.g., less than 2% on AIME2025). More importantly,~~Notably, under a fixed compute budget, our approach surpasses state-of-the-art generative verification by a significant margin: achieving up to 6.1% higher accuracy on AIME2025. Our findings establish that for practical, real-world applications, budget-aware scaling with discriminative verifiers is not only a "free" upgrade over self-consistency, but also a more effective and efficient alternative to costly generative techniques. Code is available at `https://anonymous.4open.science/r/Verification-ICLR2026`.

## 1 INTRODUCTION

The pursuit of advanced reasoning in large language models (LLMs) has been defined by the principle of scale: scaling up models, datasets, and training compute has consistently unlocked new capabilities. More recently, a new frontier has emerged in this paradigm—scaling compute not just during training, but at the point of inference. This strategy, known as test-time scaling, aims to elicit a model's full potential by allocating additional resources to solve a single problem at inference time, leading to dramatic performance gains in complex domains like mathematics and programming (OpenAI, 2024; Snell et al., 2024).

The simplest and most canonical form of test-time scaling is self-consistency (SC) (Wang et al., 2023b). Instead of trusting a single, greedily decoded answer, SC samples a diverse ensemble of solutions and selects the final answer through a simple plurality vote. This brute-force yet remarkably effective method has become a foundational baseline, demonstrating that more computation in the form of more samples often leads to better reasoning. The natural next question is whether this compute can be used more intelligently. Rather than relying on a democratic vote, could an expert "verifier" model scrutinize each solution and select the best one?

This question has given rise to a new class of powerful, state-of-the-art techniques centered on generative verification. These verifiers are themselves sophisticated LLMs that produce a detailed chain-of-thought (CoT) rationale, critically evaluating a candidate solution before rendering a final verdict (Zhang et al., 2024c; Mahan et al., 2024). The approach is intuitively appealing; it mimics human meta-cognition and opens up a new axis for scaling. If one verification pass is good, multiple passes should be even better, allowing for deeper scrutiny and higher confidence (Shi & Jin, 2025; Zhao et al., 2025).

However, this expressive power comes at a staggering computational cost. Generating a detailed CoT critique for each candidate can match or even exceed the cost of generating the original solution. This immense overhead makes generative verification impractical for many real-world applications where inference budgets are constrained. Indeed, a careful analysis by Singhi et al. (2025) reveals

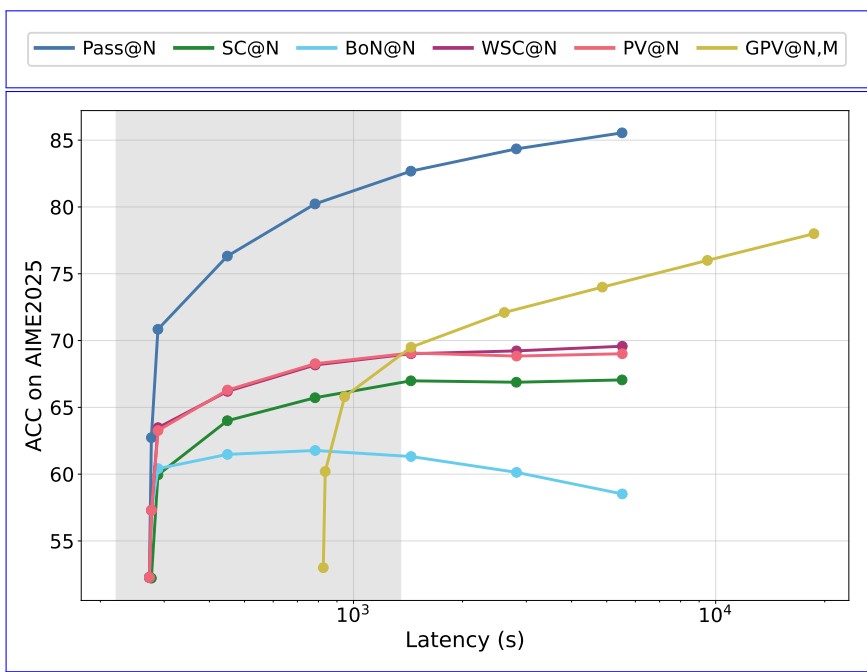

Figure 1: Hybrid discriminative verification techniques (e.g., weighted self-consistency (WSC) (Welleck et al., 2024) and pessimistic verification (PV) (Shi & Jin, 2025)) ~~consistently outperform self-consistency (SC) for a negligible amount of additional compute on AIME2025, and can~~ outperform generative pessimistic verification (GPV) under equalized compute budgets~~.~~ For latency budgetsof less than 22.5 minutes~~., discriminative verification outperforms generative verification.~~ $N$ is doubled at each point along the x-axis. For GPV, each solution is verified twice ($M = 2$). Pass@$N$ provides an upper bound on the achievable performance, assuming a free oracle verifier.

that when verification costs are properly accounted for, these state-of-the-art verification methods require up to $8\times$ more compute just to match the performance of simple self-consistency, and deliver only marginal gains even when granted a colossal $128\times$ budget.

These findings underscore an important limitation of scaling verification: solution correctness is fundamentally constrained by the quality of the candidates produced by the solver. If no correct solutions are sampled, no amount of verification, regardless of strength, can recover the right answer. Moreover, SC already provides a strong baseline, closely tracking pass@$N$ on many tasks. To improve over SC, a verifier must reliably agree with the majority when it is correct, while also identifying the minority solution when the majority is wrong. These requirements make it difficult for a verifier to deliver significant gains, especially under a fixed compute budget. As a result, allocating additional compute to generating candidate solutions typically yields better returns than spending it on verification.

Given these limitations, it is appealing to develop a budget-aware verification mechanisms that improve the model performance while minimizing compute costs. Discriminative verifiers present a promising alternative due to their computational efficiency. Unlike generative verifiers, which require both a costly prefilling step and sequential token generation during decoding, discriminative verifiers only perform a single forward pass (i.e., prefilling) to output a scalar score, thus avoiding the expensive sequential decoding bottleneck. However, despite their speed advantage, discriminative verifiers exhibit limited capabilities on complex reasoning tasks (Tan et al., 2025), often underperforming SC as the pool of candidate solutions grows, which has limited their practical use.

In this work, we show that hybrid approaches combining discriminative verification with self-consistency can offer the best trade-off between effectiveness and efficiency under practical compute budgets. ~~Although discriminative verifiers underperform SC in isolation, we show that by leveraging hybrid methods~~ (Welleck et al., 2024; Shi & Jin, 2025)~~, the resulting test-time scaling pipeline~~

~~can obtain consistent improvements over SC on AIME2025 by up to 5.1%, while having only 2% compute overhead. Moreover~~ For instance, under fixed practical inference budgets of $5 \times 10^{15}$ and $1 \times 10^{16}$ FLOPs, hybrid discriminative verification methods (Welleck et al., 2024; Shi & Jin, 2025) outperform state-of-the-art generative verification by 6.1% and 2.5%, respectively. Moreover, although discriminative verifiers underperform SC in isolation, we show that by leveraging these hybrid methods, the resulting test-time scaling pipeline can obtain consistent improvements over SC on AIME2025 by up to 5.1%, while having only 2% compute overhead. These results highlight hybrid discriminative verification as a practical and scalable alternative, delivering strong accuracy gains with negligible overhead and outperforming more expensive generative approaches under realistic budget constraints.

Our contributions are as follows:

- We conduct a thorough empirical analysis of discriminative verification techniques, exploring how different selection strategies perform across scaling regimes. To our knowledge, this is the first study to systematically examine the test-time scaling properties of discriminative verification.

- Building on this analysis, we present a compute-centric comparison of discriminative and generative verification, showing that discriminative methods offer a more practical and efficient alternative under realistic inference budgets.

## 2 EFFECTIVE DISCRIMINATIVE VERIFICATION

### 2.1 PRELIMINARIES

Repeated sampling is a test-time scaling technique that involves generating a batch of $N$ independent candidate solutions $\{s_i\}_{i=1}^N$ for a given problem $Q$. Each solution $s_i$ is a chain of reasoning that terminates in a final answer $a_i = \text{Ans}(s_i)$. As $N$ increases, the probability that at least one answer is correct also rises (i.e., Pass@$N$ improves; see Figure 1) (Cobbe et al., 2021). However, this leaves open the central challenge of selecting a single answer $a^*$ from among the candidates in the absence of ground truth.

**Self-consistency.** A common approach for this selection problem is self-consistency (SC) (Wang et al., 2023b). Since correct answers tend to recur across independent solutions, SC groups responses by their final answer and selects the most frequent one. Formally, each distinct answer $a$ has support size $n_a = |\{i : a_i = a\}|$, and SC chooses $a^* = \arg\max_a n_a$. While this approach is robust when the correct answer is common, it can fail when the majority converges on an incorrect answer. Pseudocode for this method is provided in Algorithm 1.

**Best-of-$N$.** Another strategy is best-of-$N$ (BoN) selection (Charniak & Johnson, 2005; Cobbe et al., 2021), which uses a ~~*discriminative*~~ verifier to assign each solution a scalar score (e.g., in $[0, 1]$), and selects the final answer from the highest-scoring solution. Formally, each solution $s_i$ receives a scalar score $r(s_i)$, then BoN chooses $a^* = \text{Ans}(s^*)$ where $s^* = \arg\max_{s_i} r(s_i)$. Verifiers come in two forms:

- *Discriminative verifiers* (or reward models) (Cobbe et al., 2021) output a single scalar via a value or reward head, typically using just one forward pass over the input, making them compute-efficient. In this work, BoN refers specifically to this discriminative setting.

- *Generative verifiers* (Zhang et al., 2025) prompt an LLM to judge correctness via free-form CoT reasoning. Generative verifiers can benefit from inference-time scaling by independently sampling multiple verification chains and aggregating verdicts, but incur significant compute overhead due to sequential decoding.

A strong verifier can identify correct but rare responses that SC might miss. However, as $N$ increases, it can also be misled by confident yet incorrect responses, highlighting a long-tail vulnerability (see Figure 1). Pseudocode for this method is provided in Algorithm 2.

## 2.2 Hybrid Discriminative Verification

To guard against the long-tail of high-scoring but incorrect responses, hybrid discriminative verification methods combine the consensus signal from SC with the verifier's signal from BoN. We study two hybrid approaches:

- *Weighted self-consistency* (WSC) (Welleck et al., 2024) groups solutions by their final answers and selects the answer with the largest total verifier score, i.e., $a^* = \arg\max_a \sum_{i:a_i=a} r(s_i)$. The approach prioritizes answers that are not only common but also favored by the verifier. Pseudocode for this method is provided in Algorithm 3.

- *Pessimistic verification* (PV) (Shi & Jin, 2025) groups solutions by their final answer and penalizes small answer clusters to reduce the chance of selecting low-support answers. Formally, $a^* = \arg\max_a \left( \frac{1}{n_a} \sum_{i:a_i=a} r(s_i) - \alpha \frac{\ln N}{n_a+1} \right)$, where $\alpha$ controls the strength of the penalty. When $\alpha = 0$, selection is based exclusively on the mean verifier score. As $\alpha \to \infty$, the penalty dominates and the selection collapses to SC. Empirically, we find that $\alpha = 0.5$ provides a good tradeoff (see Appendix C.1). Pseudocode for this method is provided in Algorithm 4.

## 2.3 Discriminative Verifier Training

This subsection outlines an approach for training a lightweight discriminative verifier, which provides the verification signal for BoN and hybrid discriminative verification techniques (WSC and PV).

**Dataset curation.** We sample 32k math problems from NuminaMath (LI et al., 2024), which aggregates problems from Chinese K-12 exams, Orca-Math (Mitra et al., 2024), AoPS forums, and various Olympiads (e.g., IMO, APMO, BMO), among other sources. We decontaminate the training dataset by excluding any problem whose fuzzy-match similarity to an entry in our evaluation sets exceeds 80. For each question, we sample one response from each of ten LLMs: DeepSeek-R1 and its six distilled variants (DeepSeek-AI et al., 2025), DeepScaleR-1.5B-Preview (Luo et al., 2025b), and both the preview and production releases of QWQ-32B (Team, 2024; 2025). Following Shi & Jin (2025), we remove the reasoning content (i.e., the tokens between the <think> and </think> tags) from each response (see Appendix C.2 for an ablation on this choice). Each response is graded for correctness using HuggingFace's Math-Verify toolkit (Kydlíček, 2025), which parses the model's final answer and performs symbolic equivalence checks against the reference solution. We throw out problems for which all ten solutions are either correct or incorrect, since they contain no learnable signal.

**Training.** Following prior work (Qwen et al., 2025; Yang et al., 2024), we replace the language modeling head of the LLM (specifically DeepSeek-R1-Distill-Qwen-1.5B) with a two-layer scaler value head. We train our verifier using a Bradley-Terry ranking loss combined with an $L_2$ regularization term (Ouyang et al., 2022; Kirchner et al., 2024). Concretely, our loss is

$$\mathcal{L} = -\frac{1}{|P||N|} \sum_{i \in P} \sum_{j \in N} \log \sigma(r_i - r_j) + \frac{\lambda}{2} \mathbb{E}(r^2),$$

where $r = (r_1, \ldots, r_m)$ are the logits assigned by the verifier to a batch of $m$ responses, $\sigma(x)$ is the logistic function, and $P$ and $N$ are the sets of correct and incorrect responses, respectively. The first term implements the Bradley–Terry model by maximizing the probability $\sigma(r_i - r_j)$ that every correct response $i \in P$ outranks every incorrect response $j \in N$ (Bradley & Terry, 1952), and the second term keeps score head well-behaved and centered around zero. By computing all $|P| \times |N|$ comparisons in one vectorized pass instead of sampling pairs, we gain both higher throughput and more stable gradients. We train for a single epoch on 11,420 response groups. Additional training details and hyperparameters are provided in Appendix B.

## 3 Main Results

We analyze the performance of our trained discriminative verifier under various discriminative verification techniques on several challenging benchmarks: AIME2024, AIME2025, LiveBench

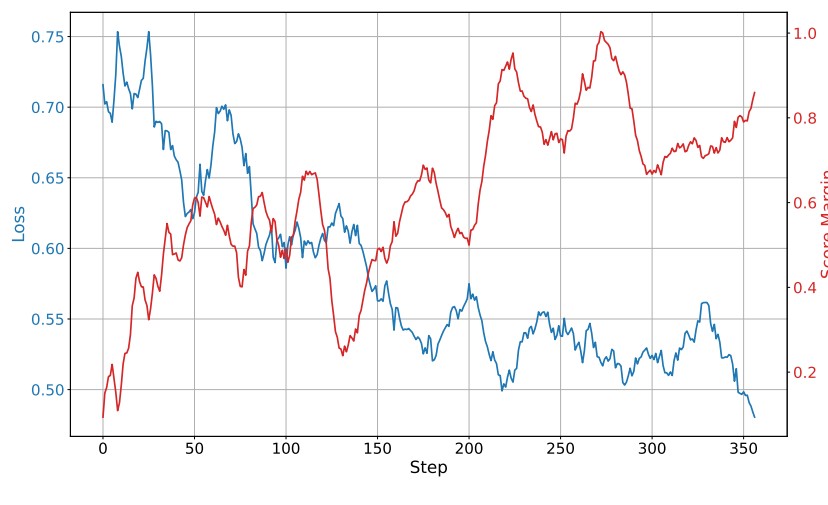

Figure 2: **Blue:** The loss decreases over one epoch of training. **Red:** The score margin, i.e., the difference in score assigned to correct solutions and incorrect solutions on average across a global batch, increases during training. Together, these indicate that the discriminative verifier learns to discriminate between correct and incorrect solutions.

Math (White et al., 2025), ~~and~~ GPQA (Rein et al., 2023), and LiveCodeBench (Jain et al., 2024). For each AIME problem, we sample 128 candidate responses no longer than 16k tokens from DeepSeek-R1-Distill-Qwen-32B. On LiveBench Math ~~and~~ GPQA, and LiveCodeBench, we sample only 64 candidate responses. Similar to the construction of our training dataset, we exclude the reasoning content (i.e., the tokens between the <think> and </think> tags) during inference (see Appendix C.2). To ensure our metric estimates (e.g., Pass@$N$ or PV@$N$) are precise, we report the mean over 1000 resampled draws of size $N$ per problem and report 95% confidence intervals. Our results are provided in Table 1.

| Method | AIME2024 | AIME2025 | LiveBench Math | GPQA | LiveCodeBench |
|---|---|---|---|---|---|
| Pass@1 | $67.0 \pm 0.5$ | $51.9 \pm 0.6$ | $62.1 \pm 0.2$ | $56.9 \pm 0.2$ | $55.6 \pm 0.4$ |
| SC@32 | $83.4 \pm 0.4$ | $66.6 \pm 0.5$ | $67.0 \pm 0.2$ | $63.5 \pm 0.2$ | $63.0 \pm 0.4$ |
| BoN@32 | $79.1 \pm 0.5$ | $60.8 \pm 0.6$ | $64.1 \pm 0.2$ | $63.9 \pm 0.2$ | $58.9 \pm 0.4$ |
| WSC@32 | $\mathbf{85.6 \pm 0.4}$ | $68.8 \pm 0.5$ | $67.5 \pm 0.2$ | $65.0 \pm 0.2$ | $\mathbf{63.5 \pm 0.4}$ |
| PV@32 | $85.5 \pm 0.4$ | $\mathbf{69.1 \pm 0.5}$ | $\mathbf{67.8 \pm 0.2}$ | $\mathbf{65.6 \pm 0.2}$ | $63.3 \pm 0.4$ |

Table 1: Accuracy rates of DeepSeek-R1-Distill-Qwen-32B ($N = 32$) with various discriminative verification techniques (highlighted in yellow). Pass@1 and SC@32 are included for comparison.

Across the board in Table 1, hybrid verification methods like WSC and PV consistently outperform competing selection methods. For example, on AIME2025, PV@32 improves over Pass@1 by 17.2%, and beats SC@32 and BoN@32 by 2.5% and 8.3%, respectively. Amazingly, even on an out-of-distribution task like GPQA, which includes questions on biology, physics, and chemistry, PV@32 can outperform SC@32 by 2.1%. On LiveCodeBench, WSC@32 and PV@32 yield smaller gains of 0.5% and 0.3% over SC@32, indicating that hybrid discriminative verification is at least as strong as SC even on out-of-distribution tasks.

### 3.1 COMPUTE-FOCUSED COMPARISON OF DISCRIMINATIVE AND GENERATIVE VERIFICATION

Recent work has explored leveraging the generative and reasoning abilities of LLMs to verify candidate solutions (Zhang et al., 2025; Mahan et al., 2024). Generative verifiers can leverage additional test-time scaling to generate and aggregate over multiple CoT rationales to produce more accurate verdicts (Zhao et al., 2025; Shi & Jin, 2025). While this strategy can boost performance, it comes at a high cost. Generative verifiers require $N(1 + M) = O(NM)$ long CoT generations per problem, where $M$ is the number of times each candidate solution is verified, leading to prohibitively

high inference costs as $N$ or $M$ is scaled. Discriminative verifiers provide a compelling alternative to generative ones: they require only a single forward pass per candidate solution, avoiding the costly decoding of long rationales. This efficiency makes them particularly attractive when compute is limited, since any budget spent on verification could otherwise be allocated to generating additional candidate solutions.

In this subsection, we compare discriminative and generative verification under equalized compute budgets. Following prior work (Singhi et al., 2025), we measure the total inference compute, i.e., the compute required to *generate and verify* candidate solutions. Concretely, we leverage Heimdall (Shi & Jin, 2025), a state-of-the-art generative verifier trained from DeepSeek-R1-Distill-Qwen-32B. Similar to hybrid discriminative verification, Heimdall leverages pessimistic verification to incorporate the consensus signal from SC, thereby improving performance. We refer to this approach as GPV (see Algorithm 5).

We focus our compute analysis on two perspectives: FLOPs and latency. FLOPs capture the theoretical compute cost of each approach, while latency reflects the real-world efficiency on modern hardware. Together, these perspectives allow us to identify the compute regimes where discriminative verifiers are most effective and where the added expense of generative verification may be justified.

### 3.1.1 FLOPS ANALYSIS

FLOPs provide a theoretical measure of the intrinsic compute required, independent of hardware and other implementation details, allowing us to study how compute requirements scale for discriminative and verification techniques. For a decoder-only transformer model with hidden size $d$, intermediate size $m$, $L$ layers, and vocabulary size $V$, the FLOPs roughly decompose into three components:

1. **Layer projections.** Each token per layer requires $8d^2 + 4dm$ FLOPs for $Q, K, V, O$ projections and the MLP.

2. **Attention.** With KV caching, prefill compute is quadratic in $T_{\text{in}}$: each of the $T_{\text{in}}$ tokens attends to all previous tokens, giving $4d \cdot \frac{T_{\text{in}}(T_{\text{in}}+1)}{2}$ FLOPs per layer. During decoding, cached keys/values avoid recomputation, so each of the $T_{\text{out}}$ generated tokens only attends to the fixed prefix and prior outputs, costing $4d \cdot (T_{\text{in}}T_{\text{out}} + \frac{T_{\text{out}}(T_{\text{out}}-1)}{2})$ FLOPs per layer.

3. **LM Head.** Finally, output projection adds $2dVT_{\text{out}}$ FLOPs, where $V$ is the vocabulary size. For discriminative verification, we set $V = 1$ and $T_{\text{out}} = 1$, corresponding to a single scalar output.

Note that this formulation omits smaller terms such as normalization layers, activation functions, or positional encodings.

We compare discriminative and generative verification methods on AIME2025. For each, we vary the number of candidate solutions $N \in 2, 4, 8, 16, 32, 64, 128$ and, for generative verification, the number of verifications per response $M \in 1, 2, 4, 8, 16, 32$. Results are presented in Figure 3.

Repeated sampling provides a natural compute baseline: generating $N$ candidate solutions requires $O(N)$ long CoT traces. For example, generating 32 candidate solutions to a problem from AIME2025 with DeepSeek-R1-Distill-Qwen-32B costs $2.0 \times 10^{16}$ FLOPs on average. SC selects the most common answer from the candidate solutions and uses no additional compute beyond that of repeated sampling. By contrast, verification-based techniques incur additional compute cost. For example, verifying 32 solutions with our discriminative verifier trained in Section 2.3 costs just $4.1 \times 10^{14}$ FLOPs on average, just 2.0% of the compute used for repeated sampling. All discriminative verification techniques (BoN, WSC, PV) use the same amount of verification compute. While BoN tends to underperform SC when $N$ is large, hybrid discriminative verification methods consistently outperform the SC baseline by up to 5.1% for a negligible amount of additional compute.

Conversely, generative verification techniques are significantly less efficient. For example, verifying the same 32 solutions with Heimdall (Shi & Jin, 2025) just once ($M = 1$) requires $3.1 \times 10^{16}$ FLOPs, over 50% more FLOPs than solution generation and nearly 76x more FLOPs than discriminative verification. While generative verification can be made more effective by scaling the number of verifications per candidate solution (i.e., increasing $M$), the compute requirements scale linearly.

Critically, under practical FLOP budgets, hybrid discriminative verification techniques outperform generative verification. This is because discriminative methods allocate nearly all of the compute

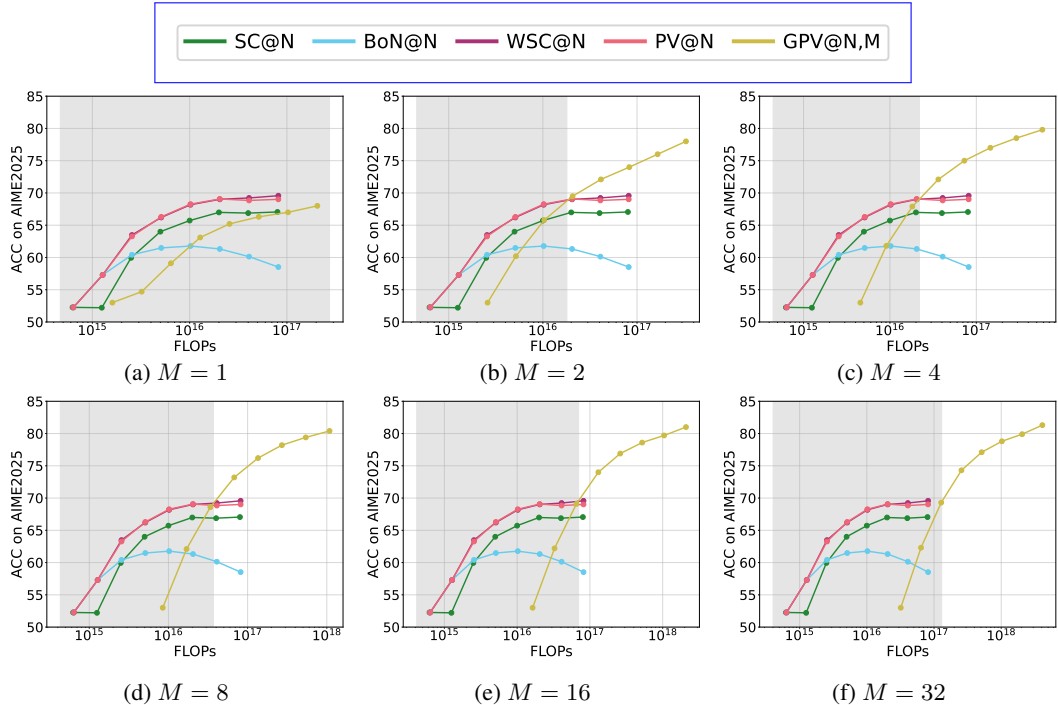

Figure 3: Accuracy vs. FLOPs on AIME2025 under equalized compute budgets. Each subplot varies the number of verifications per candidate solution $(M)$. Along each curve, successive points correspond to doubling the number of candidate solutions $(N)$. The shaded region highlights the FLOPs budgets where hybrid discriminative verification techniques strictly outperform generative verification under equalized compute budgets.

budget towards sampling candidate solutions, while generative verification splits its compute budget between sampling and verifying candidates. Under realistic compute budgets, scaling the number of candidate solutions produces greater returns than scaling verifications; even an oracle-level verifier will fail to produce the correct answer if no correct solutions were sampled. With a large enough budget, however, the gain from sampling additional candidates begins to saturate, and generative verification techniques begin to dominate. The critical threshold at which generative verification becomes superior depends on $M$ (Figure 3). For example, when $M = 1$, hybrid discriminative verification techniques outperform generative verification for any ~~combination of~~ $N \leq 128$ ~~and $M \leq 32$~~. The optimal generative configuration occurs when $M = 2$, but even still, hybrid discriminative verification methods remain optimal for compute budgets less than $2.2 \times 10^{16}$ FLOPs.

### 3.1.2 LATENCY ANALYSIS

While FLOPs provide a useful theoretical measure of compute, they do not fully capture the practical costs of inference. In real deployments, generation is often memory- and I/O-bound, with bottlenecks introduced by KV cache size, communication overhead, and sampling inefficiencies. Wall-clock latency, therefore, provides a more realistic measure of efficiency, since compute is ultimately priced in time rather than FLOPs.

We measure the average latency on AIME2025 using a single NVIDIA H100 SXM5 GPU. We leverage vLLM (Kwon et al., 2023) and its many optimizations, including dynamic batching and prefix caching, to reflect real-world usage. Similar to Section 3.1.1, we time the generation of $N \in 2, 4, 8, 16, 32, 64, 128$ candidate solutions with DeepSeek-R1-Distill-Qwen-32B and the verification of the solutions with our trained discriminative verifier and Heimdall (Shi & Jin, 2025). Latency results are reported in Table 2.

|  | $N=1$ | $N=2$ | $N=4$ | $N=8$ | $N=16$ | $N=32$ | $N=64$ | $N=128$ |
|---|---|---|---|---|---|---|---|---|
| Repeated Sampling | 273.1 | 276.6 | 288.4 | 448.4 | 782.9 | 1434.0 | 2815.5 | 5514.1 |
| Discriminative | 0.05 | 0.10 | 0.21 | 0.42 | 0.83 | 1.66 | 3.32 | 6.65 |
| Generative ($M=2$) | 552.0 | 558.8 | 656.6 | 992.8 | 1825.7 | 3423.7 | 6668.8 | 13160.7 |

Table 2: The average wall-clock time (s) for repeatedly sampling $N$ candidate solutions, as well as the average time to verify each candidate solution using discriminative and generative verification.

The latency results largely mirror the FLOP-based analysis in Section 3.1.1, but with even larger differences between discriminative and generative verification. For instance, verifying 32 solutions sampled from DeepSeek-R1-Distill-Qwen-32B with our 1.5B discriminative verifier takes only 1.66 seconds, just 0.1% of the generation time. This is an order of magnitude smaller than what FLOP estimates suggested (2.0%), reflecting the fact that discriminative verifiers avoid the decoding bottlenecks that dominate wall-clock latency.

Generative verification, by contrast, becomes even less practical under a latency perspective. Just verifying 32 candidate solutions with Heimdall at $M=2$ takes 3423.7 seconds, over twice the time needed for solution generation, and more than 2000× the cost of discriminative verification. These inefficiencies stem from the need to generate long CoTs for each verification, which incur memory-bandwidth and KV cache overheads not reflected in theoretical FLOP estimates. Indeed, as shown in Figure 1, hybrid discriminative verification methods dominate generative verification for all inference budgets shorter than 22.5 minutes (1350s) on AIME2025 with $M=2$. This threshold is dependent on a range of factors, including the number of verifications per solution $(M)$, the specific solver, the size of the verifier, and the dataset, but it highlights a broader trend: under realistic latency constraints, discriminative verification almost always gives better performance than generative verification.

In summary, while the FLOP analysis in Section 3.1.1 already showed discriminative verification to be more efficient, latency measurements make the contrast even sharper: discriminative verification achieves consistent gains for virtually the same latency as SC, whereas generative verification quickly becomes impractical as $N$ or $M$ grows.

## 3.2 SCALING MODEL SIZE FOR DISCRIMINATIVE VERIFICATION

Here, we analyze how discriminative verification techniques scale with respect to the size of the solver model, which generates the candidate solutions, and the size of the verifier, which verifies each candidate solution. To ~~do so~~study the effect of scaling the solver, we generate 128 candidate solutions per question in AIME2024 and AIME2025 using DeepSeek-R1-Distill-Qwen models with 1.5B, 7B, 14B, and 32B parameters, and verify each using our trained discriminative verifier. To isolate the effect of scaling the verifier, we train a second verifier initialized from DeepSeek-R1-Distill-Qwen-7B, and verify each candidate solution with both the 1.5B and 7B verifiers. We plot the aggregate results in Figure 4 for several values of $N$.

We observe that increasing the solver's size produces consistent but diminishing performance increases on AIME, while the effect of scaling the verifier's size is only noticeable when $N$ is sufficiently large. Specifically, hybrid methods like WSC and PV scale similarly to SC as the size of the solver is increased, with WSC and PV maintaining a consistent edge over SC regardless of the solver's size and verifier's size, across various $N$s. BoN, on the other hand, exhibits poor scaling behavior regardless of the verifier's size: when $N$ is small, BoN only slightly underperforms SC, but when $N$ is large, BoN trails far behind. These results suggest that hybrid approaches ~~can effectively mitigate~~are more effective than scaling the verifier for mitigating BoN's long-tail vulnerability.

## 3.3 INFERENCE-TIME SCALING OF DISCRIMINATIVE VERIFICATION

We study how each discriminative verification method benefits from increased inference-time compute along two axes: the number of candidate solutions sampled from the solver and the reasoning budget allocated to the solver. First, we observe that scaling $N$ produces consistent but diminishing improvements in performance on AIME (i.e., Pass@$N$ increases). BoN struggles to benefit from scaling $N$, with performance quickly saturating and even falling. On the other hand, hybrid approaches

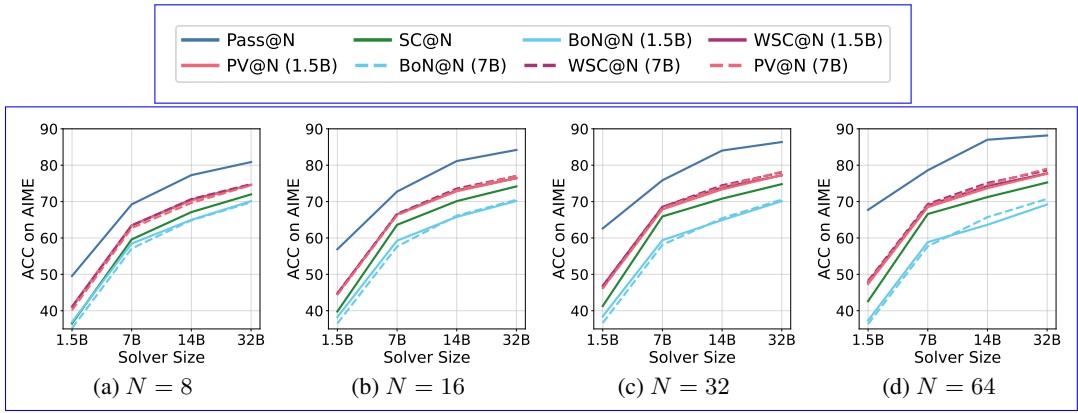

Figure 4: Accuracy rates on AIME 2024/2025 for various discriminative verification methods across four solver sizes and two verifier sizes for several values of $N$. Pass@$N$ and SC@$N$ are included as baselines.

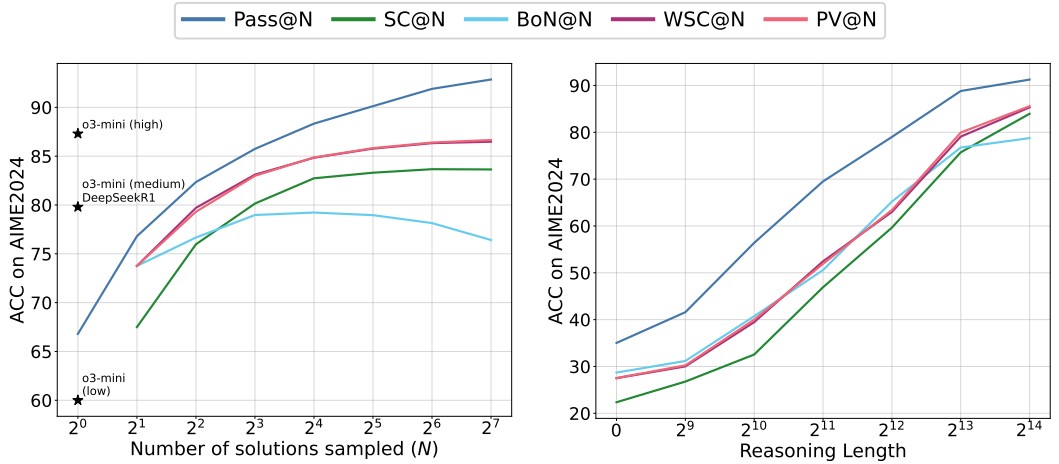

Figure 5: **Left:** Unlike BoN, hybrid techniques show consistent but diminishing improvements on AIME2024 from increasing the number of candidate results $N$ sampled from DeepSeek-R1-Distill-Qwen-32B. **Right:** The performance of DeepSeek-R1-Distill-Qwen-32B on AIME2024 scales logarithmically with the reasoning budget regardless of verification method. Here, $N = 32$.

like WSC and PV show consistent improvements as more solutions are sampled, maintaining a 2.2% to 5.6% edge over SC as $N$ is scaled from 2 to 128. ~~On AIME2024~~In Figure 5 (left), WSC and PV boost the accuracy of DeepSeek-R1-Distill-Qwen-32B on AIME2024 from $66.8\%$ to $79.7\%$ with only 4 candidate solutions, matching the performance of o3-mini (medium) or DeepSeek-R1, and outperforming SC by $3.7\%$.

To control the reasoning budget, we use budget forcing (Muennighoff et al., 2025) and truncate the candidate solutions $T \in \{0, 512, 1024, 2048, 4096, 8192, 16384\}$ tokens after the opening think tag, manually append the closing think tag, then allow the model to continue generating its final answer. In doing so, we collect solutions under constrained reasoning budgets. We observe that even as the reasoning budget is scaled from 0 to 16k tokens, WSC and PV maintain an edge over SC, even while BoN falls off (see Figure 5 (right)), showcasing the reliability of hybrid verification methods under various constraints.

## 3.4 FAILURE MODES OF DISCRIMINATIVE VERIFICATION

To better understand the limitations of hybrid discriminative verification, we analyze their failure modes on AIME2025 using 1,000 Monte Carlo trials per question. In each trial, we sample 32 candidates from the solution pool and compare WSC and PV to SC. We observe that failures are rare, with WSC and PV underperforming SC on only 2.2% and 2.6% of trials, respectively (Table 3). When failures happen, they tend to manifest in two common patterns: (1) a "minority override" error in which a high-scoring but small, incorrect cluster overrides the consensus, and

|  | WSC | PV |
|---|---|---|
| Beat SC (%) | 4.6 | 5.1 |
| Tie SC (%) | 93.2 | 92.3 |
| Lose to SC (%) | 2.2 | 2.6 |
| Minority override (%) | 69.0 | 51.3 |
| Narrow margin (%) | 31.0 | 48.7 |

Table 3: Failure modes of WSC and PV on AIME2025.

(2) a "narrow margin" error where two answers have similar frequencies but the incorrect cluster scores slightly higher. An example of the first failure mode is AIME2025 Problem 13, in which a minority cluster of size 7 (with an average score of 0.58) outscores the correct consensus cluster of size 16 (with an average score of 0.10). An example of the second failure mode is AIME2025 Problem 22, where two dominant clusters form and a narrow score margin (0.52 vs 0.47) overrides a slight consensus (41 vs 42). In practice, on problems where these patterns emerge, it may be worth turning to a stronger generative verifier to resolve the ambiguity.

## 4 RELATED WORK

**LLM Verifiers** LLM-based verifiers can be broadly categorized into generative and discriminative approaches. Generative verifiers use large language models as judges that assess the correctness or quality of outputs by generating natural language rationales. A growing body of work explores this direction, employing LLMs as judges for modeling human preferences (Dubois et al., 2024; Zheng et al., 2024; Li et al., 2024; Wang et al., 2023c; Kim et al., 2023; 2024; Li et al., 2023; Zhu et al., 2023b; Mahan et al., 2024), or as verifiers for evaluating solution correctness in reasoning tasks (Zhang et al., 2024c; Singhi et al., 2025; Shi & Jin, 2025; Saha et al., 2025).

In contrast, discriminative verifiers, such as reward models, assign scalar scores to candidate responses based on human preference data (Christiano et al., 2017; Ziegler et al., 2019; Zhu et al., 2023a; Liu & Zeng, 2024; Wang et al., 2024; Park et al., 2024; Han et al., 2024). These models are central to reinforcement learning from human feedback and are also used to rank or select responses in BoN inference settings (Lightman et al., 2023; Wang et al., 2023a; Luo et al., 2024; Saunders et al., 2022; Uesato et al., 2022; Yu et al., 2024). Together, generative and discriminative verifiers provide complementary paradigms for evaluating, selecting, and aligning LLM outputs at inference time.

**LLM Reasoning** A substantial body of work has investigated improving the mathematical reasoning capabilities of LLMs through training Cobbe et al. (2021); Guan et al. (2025); Hosseini et al. (2024); Lightman et al. (2023); Pang et al. (2024); Ye et al. (2025); Luo et al. (2025b;a), test-time scaling Snell et al. (2024); Brown et al. (2024); Setlur et al. (2024), or a combination of both Zhang et al. (2024b); Guan et al. (2025); Xie et al. (2024); Zhang et al. (2024a). Following the release of o1 OpenAI (2024), there has been a surge of interest in test-time scaling methods for LLM reasoning Snell et al. (2024); Brown et al. (2024); Singhi et al. (2025); Zhao et al. (2025), which improve performance by sampling multiple solutions and aggregating them via majority voting or LLM-based verification. Our work builds on this line of research, demonstrating that discriminative LLM verifiers can serve as an effective and efficient verification approach for test-time scaling in complex math reasoning tasks.

## 5 CONCLUSION

We studied hybrid discriminative verification as a practical alternative to costly generative approaches. Discriminative methods achieve comparable or superior accuracy in practical compute regimes, where the high cost of CoT generation limits generative approaches. Our results highlight hybrid discriminative verification as the more efficient choice for realistic test-time scaling.

REPRODUCIBILITY STATEMENT

We provide a link to our anonymized codebase in the abstract, containing everything necessary to reproduce all experiments, including the figures. In addition, we provide pseudocode for the main algorithms in Appendix A and training details and hyperparameters for our discriminative verifier in Appendix B.

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

## A  ALGORITHMS

---

**Algorithm 1** Self-Consistency (SC@$N$)

---

**Require:** problem $Q$, solver LM, slate size $N$
1: Candidates $\leftarrow \{s_i\}_{i=1}^N \sim \text{LM}(Q)$       ▷ **Stage 1: Generate Candidates**
2: Extract final answers $\{a_i\}_{i=1}^N$ and partition into clusters $\{\mathcal{C}_a\}$ by $a$   **Stage 2: Group Answers**
3: **for** each cluster $\mathcal{C}_a$ **do**
4:     $n_a \leftarrow |\mathcal{C}_a|$
5: $a^* \leftarrow \arg\max_a n_a$                ▷ **Stage 3: Plurality Vote**
6: **return** $a^*$

---

---

**Algorithm 2** Best-of-$N$ (BoN@$N$)

---

**Require:** problem $Q$, solver LM, slate size $N$, verifier $V$
  1: Candidates $\leftarrow \{s_i\}_{i=1}^N \sim \text{LM}(Q)$                     $\triangleright$ **Stage 1: Generate Candidates**
  2: Verifications $\leftarrow \{r_i = V(s_i)\}_{i=1}^N$                $\triangleright$ **Stage 2: Verify Candidates**
  3: $i^* \leftarrow \arg\max_{i \in \{1,\dots,N\}} r_i$          $\triangleright$ **Stage 3: Select Highest-Scoring Solution**
  4: $a^* \leftarrow \text{Ans}(s_{i^*})$                        $\triangleright$ **Stage 4: Extract Final Answer**
  5: **return** $a^*$

---

**Algorithm 3** Weighted Self-Consistency (WSC@$N$)

---

**Require:** problem $Q$, solver LM, slate size $N$, verifier $V$
  1: Candidates $\leftarrow \{s_i\}_{i=1}^N \sim \text{LM}(Q)$               **Stage 1: Generate Candidates**
  2: Verifications $\leftarrow \{r_i = V(s_i)\}_{i=1}^N$         $\triangleright$ **Stage 2: Verify Candidates**
  3: Extract final answers $\{a_i\}_{i=1}^N$ and partition into clusters $\{\mathcal{C}_a\}$ by $a$    **Stage 3: Group Answers**
  4: **for** each cluster $\mathcal{C}_a$ **do**
  5:      $W_a \leftarrow \sum_{i \in \mathcal{C}_a} r_i$
  6: $a^* \leftarrow \arg\max_a W_a$             **Stage 4: Select Highest-Weight Answer**
  7: **return** $a^*$

---

**Algorithm 4** Pessimistic Verification (PV@$N$)

---

**Require:** problem $Q$, solver LM, slate size $N$, verifier $V$, penalty weight $\alpha$
  1: Candidates $\leftarrow \{s_i\}_{i=1}^N \sim \text{LM}(Q)$             $\triangleright$ **Stage 1: Generate Candidates**
  2: Verifications $\leftarrow \{r_i = V(s_i)\}_{i=1}^N$         $\triangleright$ **Stage 2: Verify Candidates**
  3: Extract final answers $\{a_i\}_{i=1}^N$ and partition into clusters $\{\mathcal{C}_a\}$ by $a$    **Stage 3: Group Answers**
  4: **for** each cluster $\mathcal{C}_a$ **do**
  5:      $n_a \leftarrow |\mathcal{C}_a|$
  6:      $\bar{r}(a) \leftarrow \frac{1}{n_a} \sum_{i \in \mathcal{C}_a} r_i$
  7:      $\psi_a \leftarrow \frac{\ln N}{n_a + 1}$
  8: $a^* \leftarrow \arg\max_a \left[ \bar{r}(a) - \alpha \psi_a \right]$            $\triangleright$ **Stage 4: Select Best Answer**
  9: return $a^*$

---

**Algorithm 5** Generative Pessimistic Verification (GPV@$N, M$)

---

**Require:** problem $Q$, solver LM, slate size $N$, generative verifier $V$, # of verifications $M$, penalty weight $\alpha$
  1: Candidates $\leftarrow \{s_i\}_{i=1}^N \sim \text{LM}(Q)$             $\triangleright$ **Stage 1: Generate Candidates**
  2: **for** $i = 1$ to $N$ **do**          $\triangleright$ **Stage 2: Generative Verifications (repeat $M$ times)**
  3:      **for** $m = 1$ to $M$ **do**
  4:          $(\text{CoT}_{i,m}, r_{i,m}) \leftarrow V(s_i)$
  5:      $\tilde{r}_i \leftarrow \frac{1}{M} \sum_{m=1}^M r_{i,m}$
  6: Extract final answers $\{a_i\}_{i=1}^N$ and partition into clusters $\{\mathcal{C}_a\}$ by $a$    **Stage 3: Group Answers**
  7: **for** each cluster $\mathcal{C}_a$ **do**
  8:      $n_a \leftarrow |\mathcal{C}_a|$
  9:      $\bar{r}(a) \leftarrow \frac{1}{n_a} \sum_{i \in \mathcal{C}_a} \tilde{r}_i$
  10:      $\psi_a \leftarrow \frac{\ln(NM)}{n_a M + 1}$
  11: $a^* \leftarrow \arg\max_a \left[ \bar{r}(a) - \alpha \psi_a \right]$            $\triangleright$ **Stage 4: Select Best Answer**
  12: **return** $a^*$

---

## B   ADDITIONAL TECHNICAL DETAILS

Our training data is based on a subset of Numina-Math (LI et al., 2024). DeepSeek-R1 responses were collected from Mattern et al. (2025). Meanwhile, the majority of the responses from six DeepSeek-

R1-Distill models, DeepScaleR-1.5B-Preview, and the two QwQ models were generated on a local cluster of NVIDIA A100 GPUs, with a minority coming from 3rd party API providers.

Our evaluation datasets are AIME2024, AIME2025, LiveBench-Math (White et al., 2024), ~~and~~ GPQA (Rein et al., 2023), and LiveCodeBench Jain et al. (2024). Combined, they include ~~596~~875 questions. We decontaminate the training dataset by excluding any problem whose fuzzy-match similarity to an entry in our evaluation sets exceeds 80. For each AIME problem, we sample 128 candidate solutions, while on LiveBench Math ~~and~~, GPQA, and LiveCodeBench, we sample only 64 candidate solutions.

When rolling out solutions during training and evaluation, we follow the model's usage recommendations, namely prefilling the opening think token, sampling with a temperature of $0.6$ and a top-p value of $0.95$, and instructing the model to output its final answer within `\boxed{}`.

Our 1.5B and 7B discriminative verifiers ~~was~~were trained for a single epoch on 4xA100 SXM4 GPUs and 4xH200 SXM5 GPUs using the hyperparameters listed in Table 4.

| Hyper-parameter | Value |
|---|---|
| Global batch size | 32 |
| LR | $5 \times 10^{-5}$ |
| LR scheduler | Linear with 20 warmup steps |
| Optimizer (AdamW) | $\beta_1 = 0.9,\ \beta_2 = 0.999$ |
| $\lambda$ | 0.01 |
| Max gradient norm | 1.0 |

Table 4: Hyper-parameters for training discriminative verifiers.

## C  ADDITIONAL ABLATION EXPERIMENTS

In addition to our main experiments, we include two further ablations conducted on a held-out validation set. To construct this set, we removed 250 problems from the training dataset and generated 32 responses per problem with 1.5B, 7B, 14B, and 32B variants of deepseek-ai/DeepSeek-R1-Distill-Qwen. We discarded items where all sampled responses were correct or all incorrect, leaving 691 problems for validation. This setup ensures that both correct and incorrect responses are available, making it suitable for evaluating the performance of a verifier.

### C.1  EFFECT OF THE PESSIMISM WEIGHT $\alpha$

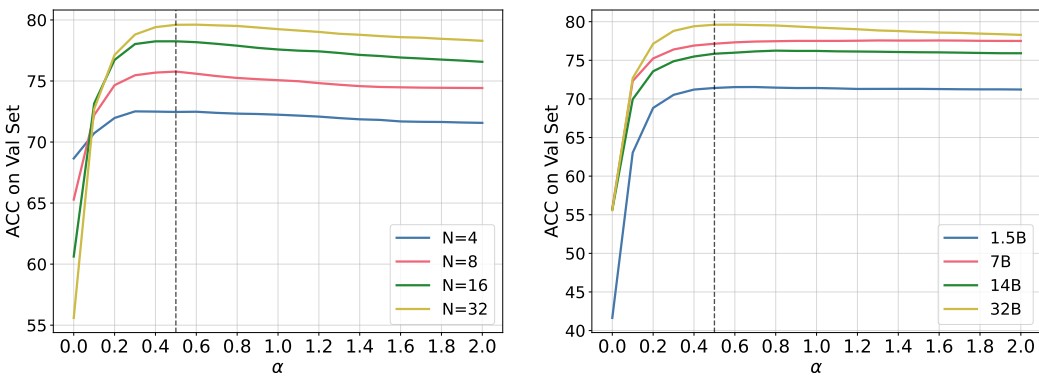

Figure 6: **Left:** Validation accuracy of PV as a function of the pessimism weight $\alpha$ for various numbers of independent candidate solutions ($N$). **Right:** Validation accuracy of PV as a function of the pessimism weight $\alpha$ for various-sized solver models.

We first ablate the effect of the pessimism weight $\alpha$ in pessimistic verification (PV). As shown in Figure 6 (left), which only includes 147 response groups generated by ~~deepseek-ai/~~DeepSeek-R1-Distill-Qwen-32B, performance peaks around $\alpha \approx 0.5$ for $N \in 4, 8, 16, 32$ and slowly decays.

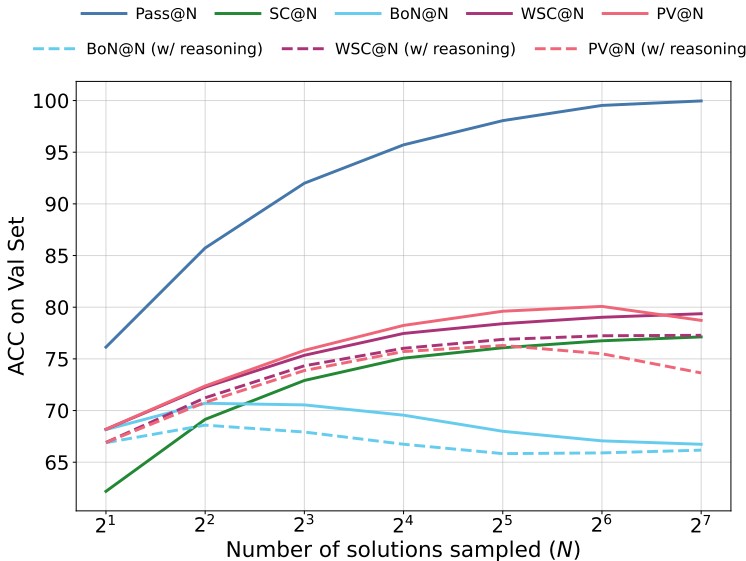

Figure 7: Validation accuracy on the held-out set when including vs. excluding reasoning content in verifier inputs for both training and inference.

Figure 6 (right) demonstrates that $\alpha = 0.5$ is a reasonable choice for 4 solver models of various sizes. Based on this result, we set $\alpha = 0.5$ for all main experiments. Moreover, we find this to be a reasonable choice for AIME2025, as well as with our 7B verifier, as the optimal $\alpha$ seems minimally dependent on either of these factors. Notably, in Shi & Jin (2025), the authors use an $\alpha = 0.1$ for experiments with Heimdall. This makes sense: with a stronger verifier and sufficiently large $M$, you can reduce $\alpha$ and put more weight on the verifier.

## C.2 EFFECT OF REASONING CONTENT ON THE VERIFIER

We next ablate whether to pass the reasoning content (the tokens between `<think>` and `</think>`) to the verifier during training and inference. Our main experiments exclude reasoning, i.e., the verifier observes only the final solution string. For comparison, we trained and evaluated a second verifier that retains the reasoning content. As shown in Figure 7, including reasoning consistently degrades performance across all selection methods: BoN, WSC, and PV all achieve lower accuracy when reasoning traces are present. This suggests that the additional reasoning text introduces noise rather than a useful signal, reinforcing our choice to exclude it during both training and evaluation.

