# OpenReview forum: "Budget-aware Test-time Scaling via Discriminative Verification"
_ICLR.cc/2026/Conference — Submitted to ICLR 2026_

### Official Review · Reviewer_Fqu6 · 2025-10-17

**Soundness:** 3
**Presentation:** 2
**Contribution:** 3
**Rating:** 8
**Confidence:** 2

**Summary:**

Work provides a detailed study of test-time scaling properties of different approaches to the verification of model responses.
Results show that a combination of discriminative and self-consistency-based methods can beat generative approaches in terms of efficiency. What is more, a verifier trained on mainly math data can bring benefits in out-of-distribution settings.

**Strengths:**

+ usage of well-established reasoning tasks, such as AIME 2024-2025 and GPQA
+ testing verification with DeepSeek R1 distilled models ranging from 1.5B to 32B parameters.
+ practical (latency) and theoretical (FLOPS) insights into test time scaling of verification methods
+ study of the influence of solver size on methods' performance
+ practical ablations, such as omission of thinking traces in verifier output

**Weaknesses:**

- Figure 1: Addition of Pass@N can significantly improve presentation, as the text suggests that Pass@N is included in the figure
```
As N increases, the probability that at least one answer is correct also rises (i.e., Pass@N improves; see Figure 1)
```
- Figure 3 can benefit from the unification of  method names with Figure 1
    - DV and BoN
    - DPV and PV
- some text inconsistencies in lines 340-342
```
For example, when M = 1, hybrid
discriminative verification techniques outperform generative verification for any combination of
N ≤ 128 and M ≤ 32.
```

**Questions:**

Can authors identify a scenario where a verifier trained on one type of data fails on the other?
Such a study can improve the contribution of the work. For example, it is unclear whether the verifier trained on math can help with code.

---

> ### Author Response · Authors · 2025-11-22
>
> We thank the reviewer for their valuable and constructive feedback. We are really happy that you enjoy this work and endorse its novel contributions. Below, we address each of your comments in detail.
>
> ### W.1 Add Pass@N to figure 1
> Great suggestion! We will add Pass@N to Figure 1 as an upper bound on the achievable performance. In this context, Pass@N reflects performance with a free (zero compute) oracle verifier.
>
> ### W.2 Unification of method names
> Thanks for catching this! We will update the legend in Figure 3 to use BoN@N and PV@N so that it aligns with the other figures. We will also double-check the writing to correct any instances of DV@N and DPV@N.
>
> ### W.3 Text inconsistencies in lines 340-342
> We will remove the "and M ≤ 32" from L341.
>
> ### Q.1 Can authors identify a scenario where a verifier trained on one type of data fails on the other?
> We primarily focus on mathematical reasoning for two reasons: (1) solutions are easily verifiable, and (2) solutions can be easily grouped by equivalence, a prerequisite for self-consistency (SC) methods. However, we do find evidence of generalization on GPQA, with WSC@32 and PV@32 outperforming SC@32 by 1.5% and 2.1%, respectively (L240-241). In other words, a verifier trained on open-ended math problems can accurately distinguish between correct and incorrect solutions to biology, physics, and chemistry problems.
>
> To see if the same is true for code generation, we evaluate these methods on LiveCodeBench [1]. We generate 64 solutions per problem with DeepSeek-R1-Distill-Qwen-32B, then group functions that produce the same outputs (or error types) on test inputs. In this setting, WSC@32 and PV@32 outperform SC@32 by 0.5% and 0.3%, respectively. Together with GPQA, this suggests that hybrid discriminative verification is at least as good as SC, even on out-of-distribution tasks. We expect that a discriminative verifier trained directly on code would deliver larger gains over SC.
>
> [1] LiveCodeBench: Holistic and Contamination Free Evaluation of Large Language Models for Code. Jain, Naman, King Han, Alex Gu, Wen-Ding Li, Fanjia Yan, Tianjun Zhang, Sida Wang, Armando Solar-Lezama, Koushik Sen, and Ion Stoica.

---

> > ### Comment · Reviewer_Fqu6 · 2025-11-25
> >
> > Thank you for the rebuttal. I have read other reviews, and I do not agree with the lack of novelty, especially as previous works have studied mostly pre-thinking/early thinking models, whereas this work studies DeepSeek-R1 distilled models and gives additional insights into the practicality of verification. Transfer of results from non-thinking models to thinking models is not obvious, and authors have noted some differences (emphasized in the rebuttal to VxsU). I maintain my positive assessment.

---

> > > ### Author Response · Authors · 2025-11-27
> > >
> > > Thank you for your follow-up and for your positive review. We appreciate your perspective and your recognition of the novelty of this work. We are committed to improving the manuscript to make our contributions clear and impactful. If you have any additional suggestions, we would be happy to include them.

---

### Official Review · Reviewer_F438 · 2025-11-01

**Soundness:** 3
**Presentation:** 2
**Contribution:** 3
**Rating:** 4
**Confidence:** 4

**Summary:**

This paper investigates budget-aware test-time scaling for LLMs via discriminative verification. While state-of-the-art generative verifiers produce detailed chain-of-thought critiques to select solutions, they incur prohibitive computational costs. The authors demonstrate that hybrid discriminative verification methods—specifically weighted self-consistency (WSC) and pessimistic verification (PV)—combine the efficiency of discriminative verifiers with the robustness of self-consistency voting. They train a 1.5B discriminative verifier using Bradley-Terry ranking loss on 32k math problems and evaluate on AIME2024/2025, LiveBench Math, and GPQA. Key findings show hybrid discriminative methods consistently outperform self-consistency with <2% overhead and surpass generative verification by up to 6.1% under fixed compute budgets (5×10¹⁵ and 1×10¹⁶ FLOPs).

**Strengths:**

1. The paper addresses a critical gap in test-time scaling research by systematically analyzing computational costs alongside accuracy.
2. Authors effectively position discriminative verification as a middle ground between pure self-consistency and expensive generative verification.
3. The authors provide multiple efficiency analysis across different metrics with both FLOP-based and latency-based compute measurements.

**Weaknesses:**

1. The evaluation focuses exclusively on mathematical reasoning tasks (AIME, LiveBench Math, GPQA). Generalizability to other reasoning domains (code generation, commonsense reasoning, strategic planning) remains unclear.
2. The paper trains a single verifier architecture (2-layer value head, Bradley-Terry loss) without comparing alternative training objectives (binary cross-entropy, contrastive learning), architectures, or verifier sizes. The choice of 0.5 for PV is validated on one held-out set but sensitivity across tasks/models is unexplored.
3. The 22.5-minute latency and 2.2×10^16 FLOPs crossover thresholds are specific to the experimental setup (DeepSeek-R1-Distill-Qwen-32B solver, Heimdall with M=2). No analysis examines failure modes where hybrid discriminative methods underperform self-consistency, limiting practical guidance for deployment.

**Questions:**

See weaknesses.

---

> ### Author Response · Authors · 2025-11-22
>
> We thank the reviewer for their insightful comments and for recognizing the contributions of this work on addressing a critical gap in test-time scaling research. We agree that this is an urgent and impactful direction, and we are committed to addressing your comments below to further strengthen this work.
>
> ### W.1 Generalizability to other domains remains unclear
> We primarily focus on mathematical reasoning for two reasons: (1) solutions are easily verifiable, and (2) solutions can be easily grouped by equivalence, a prerequisite for self-consistency (SC) methods. However, we do evaluate on GPQA, which is an out-of-distribution task containing multiple-choice questions about biology, physics, and chemistry, and find evidence of generalization, with WSC@32 and PV@32 outperforming SC@32 by 1.5% and 2.1%, respectively (L240-241).
>
> Moreover, as per your suggestion, we evaluate these methods on LiveCodeBench [1]. We generate 64 solutions per problem with DeepSeek-R1-Distill-Qwen-32B, then group functions that produce the same outputs (or error types) on test inputs. In this setting, WSC@32 and PV@32 outperform SC@32 by 0.5% and 0.3%, respectively. Together with GPQA, this suggests that hybrid discriminative verification is at least as good as SC, even on out-of-distribution tasks. We expect that a discriminative verifier trained directly on code would deliver larger gains over SC.
>
> ### W.2 Limited verifier architecture and hyperparameter exploration
> To directly address this concern, we trained a second verifier based on DeepSeek-R1-Distill-Qwen-7B, following the same methodology as for the 1.5B verifier. The results (mirroring Table 1) are below:
>
> |        |AIME2024|AIME2025|LiveBench Math|GPQA|
> |--------|--------|--------|--------------|----|
> |Pass@1  |67.0    |51.9    |62.1          |56.9|
> |SC@32   |83.4    |66.6    |67.0          |63.5|
> |BoN@32  |79.0    |62.2    |67.4          |65.1|
> |WSC@32  |86.3    |70.1    |67.5          |66.1|
> |PV@32   |86.4    |70.3    |68.3          |66.3|
>
> On all four tasks, WSC@32 and PV@32 with the 7B verifier perform at least as well as with the 1.5B verifier. For example, on AIME2025, WSC@32 and PV@32 improve by 1.3% and 1.2%, respectively. In addition, the FLOPs crossover point at which generative verification begins to outperform discriminative methods shifts from 2.2x10^16 to 2.7x10^16 FLOPs, indicating that end-to-end performance improves as the verifier is scaled.
>
> On a held-out validation set, we vary (1) the number of candidate solutions N, (2) the solver size, and (3) the verifier size (1.5B vs 7B). Across these axes, we find that performance is largely stable for $\alpha \in [0.4, 0.8]$. When $\alpha < 0.4$, accuracy increases sharply with $\alpha$, and when $\alpha > 0.8$, accuracy decays only gradually. Running the same analysis on AIME2025 yields similar trends, confirming $\alpha = 0.5$ as a good choice across experimental settings. We will update Appendix C.1 to include these new AIME2025 and 7B verifier results.
>
> ### W.3 Crossover thresholds are setup-specific and missing analysis of failure modes
> The crossover thresholds are specific to the experimental setup. For example, increasing the discriminative verifier from 1.5B to 7B parameters shifts this crossover point from 2.2x10^16 to 2.7x10^16 FLOPs (with DeepSeek-R1-Distill-Qwen-32B as the solver and Heimdall as the generative verifier with M=2). The key insight of this work is that every practical configuration has such a crossover threshold, and this threshold often exceeds most practical test-time scaling budgets.
>
> Additionally, we analyze failure modes on AIME2025 and observe that WSC/PV underperform SC on only 6-10% of instances. Failures manifest in two patterns: (1) a high-scoring but small, incorrect cluster overrides the consensus, and (2) two answers have similar frequencies but the incorrect cluster scores slightly higher. An example of the first failure mode is AIME2025 Problem 13, in which a minority cluster of size 7 (avg score of 0.58) outscores the correct consensus cluster with size 16 (avg score of 0.10). An example of the second failure mode is AIME2025 Problem 22, where two dominant clusters form and a narrow score margin (0.52 vs 0.47) overrides a narrow consensus (41 vs 42). We will incorporate this discussion into the manuscript.
>
> [1] LiveCodeBench: Holistic and Contamination Free Evaluation of Large Language Models for Code. Jain, Naman, King Han, Alex Gu, Wen-Ding Li, Fanjia Yan, Tianjun Zhang, Sida Wang, Armando Solar-Lezama, Koushik Sen, and Ion Stoica.

---

### Official Review · Reviewer_VxsU · 2025-11-01

**Soundness:** 2
**Presentation:** 2
**Contribution:** 1
**Rating:** 2
**Confidence:** 4

**Summary:**

This paper analyzes the test-time scaling properties of discriminative verifiers with some commonly used selection techniques, like Best-of-N, weighted self-consistency, etc. They show that using techniques like weighted self-consistency with discriminative verifiers is a good way to improve performance over self-consistency (without a verifier) while not being as computationally expensive as using generative verifiers, which have been shown to be very expensive by prior work.

**Strengths:**

This paper compared various approaches (like best-of-N, weighted SC, etc.) as well as compared discriminative and generative verifiers. It compared them in terms of performance, computational cost (both FLOPs and latency), and scaling properties along various axes like the number of candidate solutions, the length of each solution, etc.

**Weaknesses:**

The major weakness is that most of this is already well-known and not really novel.
1. One of the primary results in the paper is that with discriminative verifiers, weighted self-consistency or GPV > self-consistency. This is pretty well-known (for example, figure 6 in [1], figure 7 in [2], figure 8 in [3], figure 1 in [4]; also see [5]).
1. The finding that discriminative verifiers can output the simple self-consistency baseline is also not new (see figure 5 in [2]).
1. The finding that discriminative verifiers are less expensive than generative verifiers is also quite expected since they don’t generate verification tokens at inference time. In terms of inference cost, they would be similar to majority voting.
1. The findings in Sections 3.2 and 3.3 are also quite expected: if the candidate solutions improve in performance, then the overall performance would increase. For example, Figure 5a is qualitatively similar to Figure 7 in [2].

## A few minor nit-picks:
- Preliminaries don’t explain what (discriminative) verifiers are.
- The Section title “Scaling Model Size for Discriminative Verification” was a bit misleading because it seemed like the focus would be on scaling verifier size (which wasn’t the case). It might be better to rephrase this for clarity.
- Section 3.3 paragraph 1 – refer to a figure or table supporting these numbers (i.e., figure 5).


## References
[1] From Decoding to Meta-Generation: Inference-time Algorithms for Large Language Models. Sean Welleck, Amanda Bertsch, Matthew Finlayson, Hailey Schoelkopf, Alex Xie, Graham Neubig, Ilia Kulikov, Zaid Harchaoui.

[2] Large Language Monkeys: Scaling Inference Compute with Repeated Sampling. Bradley Brown, Jordan Juravsky, Ryan Ehrlich, Ronald Clark, Quoc V. Le, Christopher Ré, Azalia Mirhoseini.

[3] Generative Verifiers: Reward Modeling as Next-Token Prediction. Lunjun Zhang, Arian Hosseini, Hritik Bansal, Mehran Kazemi, Aviral Kumar, Rishabh Agarwal.

[4] Scaling LLM Test-Time Compute Optimally can be More Effective than Scaling Model Parameters. Charlie Snell, Jaehoon Lee, Kelvin Xu, Aviral Kumar.

[5] Improving Large Language Model Fine-tuning for Solving Math Problems. Yixin Liu, Avi Singh, C. Daniel Freeman, John D. Co-Reyes, Peter J. Liu.

**Questions:**

No major questions

---

> ### Author Response · Authors · 2025-11-22
> **Official Comment by Authors (1/2)**
>
> We thank the reviewer for their thoughtful and constructive comments. We recognize the main concern of your review centers on the novelty of our results. We believe this stems from a misunderstanding of our contributions and of existing works. Our primary contribution is showing that **discriminative verifiers can offer a more practical and efficient alternative to generative verifiers under realistic inference budgets (L118-119)**. Existing works, by contrast, either demonstrate that generative verifiers outperform discriminative verifiers [3] or that discriminative verifiers can augment self-consistency [1, 2]. Our findings are therefore both novel and practically important: many academic and industry deployments operate under tight test-time budgets and may rely on suboptimal configurations. In the few places where our results touch on prior trends (e.g., Section 3.2), it is precisely because the literature lacks a clear consensus, and our analysis seeks to resolve this ambiguity. Below, we address each concern in turn.
>
> ### W.1 Discriminative verifiers + WSC/PV > SC is well-known
> Our contribution is not merely showing that discriminative verification can outperform SC, but that it does so consistently and at negligible compute cost, neither of which is established in prior work.
>
> First, prior work on weighted self-consistency (WSC) lacks a clear consensus. For example, Figure 7 in [2] and Figure 8 in [3] show the gap between SC and WSC narrowing, even disappearing, as the number of candidate solutions increases. By contrast, in Figure 5 from our work (and Figure 6 in [1]), this gap remains consistent. In addition, as far as we are aware, no prior work has studied discriminative verification with an explicit pessimism penalty (our PV setting), which we show performs comparably to WSC.
>
> Second, we evaluate on substantially harder problems (e.g., AIME) using much weaker verifiers. For instance, DeepSeek-R1-Distill-Qwen-32B scores 94.3% on MATH-500 [6] but only 51.9% on AIME2025. In such a regime, it is far from obvious that a 1.5B verifier should be effective at all, yet we show that, when used appropriately, it still yields clear and meaningful improvements.
>
> Finally, unlike prior work, we explicitly frame the comparison in terms of compute allocation. We show that even a small 1.5B parameter Bradley-Terry verifier yields consistent gains over SC for a negligible compute overhead (e.g., <2%). Prior work, in contrast, requires larger and more complex verifiers to surpass SC. For example, [1] uses a 34B-parameter process reward model (~20x larger), [2] uses an 8B multi-objective reward model with mixture-of-experts gating, and [5] explicitly notes that "only the PaLM 2-L evaluator can outperform the robust majority-voting baseline" and that "re-ranking is a difficult task for relatively smaller models". In these cases, beating SC comes at a non-trivial compute cost, making discriminative verification far less attractive in practice. By contrast, our work is the first to show that even a weak verifier can beat SC with negligible compute overhead.
>
> ### W.2 Discriminative verifiers > SC is not new
> We do not claim that discriminative verification by itself (i.e., without the SC signal) outperforms SC; in fact, we observe the exact opposite (L099-101, L143-145, L413-415, and L421-422), which motivates our study of hybrid techniques (i.e., WSC and PV).
>
> However, the existing literature on this point is mixed and far from conclusive. For instance, when the number of candidate solutions is sufficiently large:
> - Figure 6 in [1] shows BoN and SC converging at the same performance on MATH
> - Figure 7 in [2] shows BoN underperforming SC on GSM8K and MATH
> - Figure 5 in [3] shows BoN outperforming SC on GSM8K, but underperforming on MATH
> - Figure 1 in [4] shows BoN consistently outperforming SC on MATH
>
> Our results most closely resemble [2]: BoN can outperform SC when N is small (e.g., N = 4), but falls below SC as N grows (Figure 5, left). We attribute this to a long-tail failure mode of discriminative verifiers, where increasing N increases the likelihood of selecting a plausible but incorrect solution.
>
> Crucially, in our experiments, discriminative verification only reliably outperforms SC when its signal is combined with SC, such as through WSC or PV. The consistent gains achieved by these hybrid strategies are not established in prior work.

---

> > ### Author Response · Authors · 2025-11-22
> > **Official Comment by Authors (2/2)**
> >
> > ### W.3 Discriminative verifiers are obviously cheaper than generative verifiers
> > We agree that discriminative verification is cheaper than generative verification in principle. Section 3.1, however, makes a stronger point: **the cost gap is large enough to fundamentally change the optimal compute allocation strategy.**
> >
> > In realistic regimes, performance is constrained by the solver’s ability to generate at least one correct candidate. No verifier, generative or discriminative, can fix a batch of uniformly wrong solutions. Thus, under tight compute budgets, verification cost must be minimized so that more budget can be allocated to sampling.
> >
> > Under equalized compute, discriminative verification combined with SC (i.e., WSC or PV) delivers consistent, near-free gains over SC and even outperforms generative verifiers. To our knowledge, no prior work demonstrates either phenomenon.
> >
> > ### W.4 The findings in Sections 3.2 and 3.3 are expected
> > The goal of Sections 3.2 and 3.3 is to show that techniques like WSC and PV perform well in regimes not covered by Table 1. Even if some qualitative trends are intuitive (e.g., better solver -> better performance), our contribution is to quantify these effects and characterize how they scale. These results are not implied by prior work, including Figure 7 in [2], for several reasons:
> >
> > 1. Our findings are not quantitatively similar to [2]. For instance, while [2] shows WSC and SC converging at the same level of performance, our Figure 4 shows consistent improvements over SC with both WSC and PV.
> > 2. Our setting is substantially more difficult. We evaluate on much harder problems (e.g., AIME), use a weaker verifier (~5x smaller than [2]) and lack MoE gating, yet still achieve strong performance.
> > 3. We are, to our knowledge, the first to show that discriminative verification methods such as WSC and PV can match the performance of state-of-the-art verifiers like o3.
> > 4. Prior work focuses solely on increasing the number of samples. We additionally analyze longer sampled reasoning, a regime that, to our knowledge, has not been explored previously.
> >
> > ### Minor nit-picks
> > We appreciate the comments and are preparing a revision that takes these into account.
> >
> > [1] From Decoding to Meta-Generation: Inference-time Algorithms for Large Language Models. Sean Welleck, Amanda Bertsch, Matthew Finlayson, Hailey Schoelkopf, Alex Xie, Graham Neubig, Ilia Kulikov, Zaid Harchaoui.
> >
> > [2] Large Language Monkeys: Scaling Inference Compute with Repeated Sampling. Bradley Brown, Jordan Juravsky, Ryan Ehrlich, Ronald Clark, Quoc V. Le, Christopher Ré, Azalia Mirhoseini.
> >
> > [3] Generative Verifiers: Reward Modeling as Next-Token Prediction. Lunjun Zhang, Arian Hosseini, Hritik Bansal, Mehran Kazemi, Aviral Kumar, Rishabh Agarwal.
> >
> > [4] Scaling LLM Test-Time Compute Optimally can be More Effective than Scaling Model Parameters. Charlie Snell, Jaehoon Lee, Kelvin Xu, Aviral Kumar.
> >
> > [5] Improving Large Language Model Fine-tuning for Solving Math Problems. Yixin Liu, Avi Singh, C. Daniel Freeman, John D. Co-Reyes, Peter J. Liu.
> >
> > [6] DeepSeek-R1: Incentivizing Reasoning Capability in LLMs via Reinforcement Learning. DeepSeek-AI.

---

### Author Response · Authors · 2025-11-26
**Summary of Revisions**

We have updated the manuscript with a revision taking into account the feedback we received from the reviewers. For your convenience, new content is highlighted in blue, and removed text is struck through in red. Below, we summarize our revisions:
- `VxsU` raised concerns regarding the novelty of our work. To clarify this, we refined parts of the abstract (L17-20) and introduction (L77-80, L107-114) to emphasize our contributions related to the compute-centric comparison of discriminative and generative verification.
- `VxsU` noted that the preliminaries did not clearly define discriminative verifiers. In response, we added a discussion introducing and comparing discriminative and generative verifiers to Section 2.1 (L144-152).
- `VxsU` found the scaling section misleading, and `F438` questioned the limited verifier architecture. To address this, we trained a second, larger verifier from DeepSeek-R1-Distill-Qwen-7B. As such, we updated Section 3.2 to focus on scaling the verifier's size, in addition to the solver's size.
- `VxsU` requested a clearer reference to supporting figures in Section 3.3. To make this explicit, we updated Section 3.3 (L466 and L475) to explicitly refer to the relevant figures (i.e., Figure 5, left or right).
- `F438` and `Fqu6` raised questions about generalization beyond mathematical reasoning. To investigate this, we added LiveCodeBench as a second out-of-distribution task (in addition to GPQA), and updated Table 1 as well as the corresponding discussion in L255-257.
- `F438` wanted more analysis of $\alpha$ in pessamistic verification. To better understand its effect, we extended Appendix C.1 to include a discussion around the connection between the optimal $\alpha$ and the verifier's size and the evaluation dataset.
- `F438` asked for insight into failure modes. To address this, we extended Section 3 with a subsection on failure modes, highlighting and categorizing cases where hybrid methods underperform self-consistency.
- `Fqu6` suggested including Pass@N in Figure 1. In line with this suggestion, we added Pass@N to Figure 1 to serve as an upper bound on the achievable performance, assuming a free oracle verifier.
- `Fqu6` noted inconsistent method naming across figures. To fix this, we corrected the legend in Figure 3 and confirmed that the terminology is consistent throughout the manuscript.
- `Fqu6` identified an inconsistency in L356 (formerly L341). To correct this, we removed the "and M ≤ 32" from L356.

We again express our sincerest appreciation for the valuable and constructive comments we received from the reviewers. We hope to continue engaging with the reviewers during this discussion period, as we are committed to further strengthening this work.

---

### Author Response · Authors · 2025-12-03
**Summary of Reviewers' Feedback and Our Rebuttals**

While it is unfortunate that the discussion period between authors and reviewers was cut short, we welcome the discretionary attention from the AC. For the convenience of the AC and the general public, we provide a summary of our work, the feedback we received, and our rebuttals.

### TL;DR of Our Work
Our work investigates how to best allocate compute for test-time scaling, focusing on discriminative and generative verification techniques. We found that the immense overhead of generative verification limits its practicality, often requiring impractical compute budgets to see gains over self-consistency.

To address this, we turn to a more budget-friendly paradigm: discriminative verification. While discriminitive verification performs poorly, hybrid approaches (e.g., weighted self-consistency (WSC) or pessimistic verification (PV)) are remarkably effective for their cost. Under equalized compute budgets, hybrid discriminative verification can outperform generative verification by up to 6.1% on AIME2025.

In short, our primary contribution is the compute-driven comparison of generative and discriminative verification methods, showing that
discriminative verifiers can offer a more practical and efficient alternative to generative verifiers under realistic inference budgets. Secondly, to resolve ambiguities in prior work, we provide a systematic scaling analysis of discriminitive verification techniques focusing on the size of the solver, the size of the verifier, the number of candidate solutions, and the solver's reasoning budget.

### Reviewers' Recognition
Below, we highlight several of the positive comments our work received from reviewers.

**Reviewers find that our work addresses a critical and practical gap in test-time scaling research.**
- `F438`: *"The paper addresses a critical gap in test-time scaling research by systematically analyzing computational costs alongside accuracy."*
- `Fqu6`: *"Practical ... insights into test time scaling of verification methods"*

**Reviewers agree that hybrid discriminative approaches effectively balance efficiency and robustness.**
- `F438`: *"Authors effectively position discriminative verification as a middle ground between pure self-consistency and expensive generative verification."*
- `Fqu6`: *"Results show that a combination of discriminative and self-consistency-based methods can beat generative approaches in terms of efficiency."*

**Reviewers appreciate the rigor, breadth, and scale of our experimental analysis.**
- `VxsU`: *"This paper compared various approaches... [and] compared them in terms of performance, computational cost (both FLOPs and latency), and scaling properties along various axes like the number of candidate solutions, the length of each solution, etc."*
- `F438`: *"The authors provide multiple efficiency analysis across different metrics with both FLOP-based and latency-based compute measurements."*
- `Fqu6`: *"[The authors give] practical (latency) and theoretical (FLOPS) insights into test time scaling of verification methods [and] practical ablations, such as omission of thinking traces in verifier output"*

---

> ### Author Response · Authors · 2025-12-03
> **Summary of Reviewers' Feedback and Our Rebuttals**
>
> ### Reviewers' Concerns and Our Rebuttals
> Other than some presentation suggestions, the reviewers raise the following concerns about our work:
>
> **`VxsU` raised concerns regarding the novelty of our work -- addressed.**
> - We clarified that our primary contribution is the compute-centric comparison demonstrating that discriminative verifiers can outperform generative verifiers under practical inference budgets. In places where our scaling analysis overlaps with prior work, it is because the prior work is inconclusive, or our results cannot be inferred from the prior work. Reviewer `Fqu6` agrees, recognizing the novelty of our work.
> - To address this, we refined parts of the abstract (L17-20) and introduction (L77-80, L107-114) to emphasize our contributions related to the compute-centric comparison of discriminative and generative verification.
>
> **`VxsU` found the model scaling section misleading, and `F438` questioned the limited verifier architecture -- addressed.**
> - We conducted a new experiment and trained a 7B discriminitive verifier from DeepSeek-R1-Distill-Qwen-7B. We observe that WSC@32 and PV@32 with the 7B verifier perform at least as well as with the 1.5B verifier. For example, on AIME2025, WSC@32 and PV@32 improve by 1.3% and 1.2%, respectively.
> - To address this, we updated Section 3.2 with results from the 7B verifier on AIME2025. Now, the section focuses on scaling the verifier's size, in addition to the solver's size.
>
> **`F438` and `Fqu6` raised questions about generalization beyond mathematical reasoning -- done.**
> - We observed some degree of generalization to out-of-distribution tasks (i.e., beyond mathematical reasoning). For example, on GPQA (biology, physics, and chemistry), WSC@32 and PV@32 outperform SC@32 by 1.5% and 2.1%, respectively. We conducted an additional experiment on LiveCodeBench to measure generalization to code generation tasks. In this setting, we find weak generalization, with WSC@32 and PV@32 only outperforming SC@32 by 0.5% and 0.3%, respectively. Together, this suggests that hybrid discriminative verification is at least as good as SC, even on out-of-distribution tasks.
> - We updated the manuscript to include the new results on LiveCodeBench. Specifically, we updated Table 1 and the corresponding discussion in L255-257.
>
> **`F438` wanted more analysis of $\alpha$ in pessamistic verification -- added.**
> - We provided analysis showing that performance is largely stable for $\alpha \in [0.4, 0.8]$. We confirm this along three dimensions on a held-out validation set: the number of candidate solutions, the size of the solver, and the size of the verifier. At `F438`'s request, we repeat this analysis on AIME2025, and observe that the same trends hold.
> - We extended Appendix C.1 to include a discussion around the connection between the optimal $\alpha$ and the verifier's size and the evaluation dataset.
>
> **`F438` asked for insight into failure modes -- added.**
> - We conducted a thorough analysis of the failure modes of WSC and PV on AIME2025. We identified that while hybrid methods rarely underperform SC, their errors manifest as either "minority override" or "narrow margin" errors.
> - We added a new subsection (Section 3.4) on failure modes to the manuscript (L485-502) to highlight these findings.
>
> While it is unfortunate that the discussion period was cut short before reviewers `VxsU` and `F438` were able to respond, we firmly believe that our rebuttals have faithfully addressed the weaknesses they raised and hope that the AC recognizes this. Also, we have prepared and uploaded a revision that incorporates these changes. For your convenience, new content is highlighted in blue, and removed text is struck through in red.

---

### Meta-Review · Area_Chair_mqx2 · 2026-01-06

**Summary:**

The paper studies budget-aware test-time scaling for LLM reasoning using discriminative verifiers, and proposes hybrid selection (WSC/PV) that mixes verifier scores with self-consistency. Strengths are the compute-centric framing, broad scaling sweeps (solver/verifier size, N, budgets), and both FLOPs and latency reporting. Main weaknesses are limited novelty (largely known selection recipes plus careful measurement), and limited coverage beyond settings where answer clustering is easy.

**Reviewer Concerns:**

The rebuttal addresses several actionable issues: clearer definitions/positioning, corrected presentation issues, added OOD evaluation (LiveCodeBench), added a larger 7B discriminative verifier experiment, and added sensitivity/failure-mode analysis for PV/WSC. The main outstanding concern is still novelty relative to prior work on WSC-style hybrids and verification scaling, plus limited generality beyond tasks with well-defined equivalence grouping.

**Reviewer Scores:**

VxsU (2 reject) would likely stay a reject because the core objection is novelty.

F438 (4 borderline reject) might move slightly upward given added experiments (7B verifier, code benchmark, failure modes), but could remain below threshold due to limited breadth.

Fqu6 (8 accept) would likely stay accept and might increase confidence a bit after the rebuttal and cross-review discussion.

---

### Decision · Program_Chairs · 2026-01-26

Reject